# Morphological Electrical and Hardness Characterization of Carbon Nanotube-Reinforced Thermoplastic Polyurethane (TPU) Nanocomposite Plates

**DOI:** 10.3390/molecules28083598

**Published:** 2023-04-20

**Authors:** José Muñoz-Chilito, José A. Lara-Ramos, Lorena Marín, Fiderman Machuca-Martínez, Juan P. Correa-Aguirre, Miguel A. Hidalgo-Salazar, Serafín García-Navarro, Luis Roca-Blay, Luis A. Rodríguez, Edgar Mosquera-Vargas, Jesús E. Diosa

**Affiliations:** 1Grupo de Transiciones de Fase y Materiales Funcionales, Departamento de Física, Universidad del Valle, Santiago de Cali 760032, Colombia; 2Centro de Excelencia en Nuevos Materiales (CENM), Universidad del Valle, Santiago de Cali 760032, Colombia; 3Grupo de Películas Delgadas, Universidad del Valle, Santiago de Cali 760032, Colombia; 4Grupo de Investigación en Procesos Avanzados para Tratamientos Biológicos y Químicos, Escuela de Ingeniería Química, Universidad del Valle, Santiago de Cali 760032, Colombia; 5Grupo de Investigación en Tecnología para la Manufactura, Universidad Autónoma de Occidente, Santiago de Cali 760035, Colombia; 6AIMPLAS, Gustave Eiffel 4 (València Parc Tecnològic), 46980 Paterna, Spain

**Keywords:** thermoplastic polyurethane, multi-walled carbon nanotubes, nanomaterial-reinforced polymer, impedance spectroscopy, equivalent circuit modeling, percolation conduction

## Abstract

The impacts on the morphological, electrical and hardness properties of thermoplastic polyurethane (TPU) plates using multi-walled carbon nanotubes (MWCNTs) as reinforcing fillers have been investigated, using MWCNT loadings between 1 and 7 wt%. Plates of the TPU/MWCNT nanocomposites were fabricated by compression molding from extruded pellets. An X-ray diffraction analysis showed that the incorporation of MWCNTs into the TPU polymer matrix increases the ordered range of the soft and hard segments. SEM images revealed that the fabrication route used here helped to obtain TPU/MWCNT nanocomposites with a uniform dispersion of the nanotubes inside the TPU matrix and promoted the creation of a conductive network that favors the electronic conduction of the composite. The potential of the impedance spectroscopy technique has been used to determine that the TPU/MWCNT plates exhibited two conduction mechanisms, percolation and tunneling conduction of electrons, and their conductivity values increase as the MWCNT loading increases. Finally, although the fabrication route induced a hardness reduction with respect to the pure TPU, the addition of MWCNT increased the Shore A hardness behavior of the TPU plates.

## 1. Introduction

Interesting applications have been proposed for nanocomposite materials consisting of well-dispersed carbon nanotubes (CNTs) into a polymer matrix. The inclusion of CNTs helps to improve the electrical conductivity and hardness of the polymer. A conducting and flexible character of such nanocomposites can be used in the development of high-performance sensors [1] that could be modulated by strain [2,3,4], generating a flexible strain sensor with multiple applications in the field of biomedical engineering [4], as a piezoresistive sensor with ultrasensitive detection [5,6,7] and as an electromagnetic interference shielding device [8], among others.

As a polymer matrix, thermoplastic polyurethane (TPU) is an excellent candidate [9]. It has many desirable properties such as high elasticity and durability; abrasion resistance; good flexibility over a wide range of temperature; and resistance to oils, greases, and numerous solvents [10,11]; it is also an easy processable and recyclable material [12,13,14]. From an electrical point of view, the insulating or dielectric character of TPU can be drastically modified with the inclusion of conductive fillers, or by the formation of a gel polymer electrolyte, that promotes the electrical conductivity of the system by electronic percolation [6,15,16,17,18,19,20,21] or ion conduction [22,23,24,25,26].

The use of carbon-based nanostructures such as single-walled carbon nanotubes (SWCNTs) and multi-walled carbon nanotubes (MWCNTs) as fillers that promote the reinforcement and electrical conductivity of TPU has been intensively studied, both experimentally and numerically [6,8,16,17,19,20,21,27,28,29,30,31,32,33]. It has been reported that the inclusion of a low content of SWCNTs or MWCNTs significantly enhances the hardness and the electrical conductivity of the polymer. However, in the case of the electrical studies, most of them are focused on measuring the total electrical conductivity of the nanocomposite, which is mainly mediated by an electronic percolation process through conductive pathways generated by carbon nanotubes. Other conduction mechanisms could be detected if we use a powerful electrical characterization technique such as impedance spectroscopy in combination with equivalent circuit modeling [34,35].

In this work, we perform a morphological, crystalline, electrical and hardness investigation of TPU/MWCNT plates fabricated by compression molding from nanocomposite pellets obtained by an extrusion process. This manufacturing route has been widely used for the production of these nanocomposites [15,20,21,22,35,36]; the extrusion process ensures good dispersion of the embedded CNTs, and the compression molding enables us to fabricate two-dimensional nanocomposites, a convenient form to perform mechanical tests and to develop flexible strain sensors or electromagnetic interference shielding [3,4,8,16,21,30,37]. The MWCNT content varied from 1 to 7 wt%, an interesting MWCNT concentration range that should be above the percolation threshold and where numerous investigations have been carried out with the aim of tuning the electrical and mechanical properties of this nanocomposite. In the case of electrical characterization, a detailed impedance spectroscopy analysis was performed.

## 2. Results and Discussion

### 2.1. Morphological and Crystalline Characterization

As a representative sample, Figure 1 displays the cross-sectional cryo-SEM images recorded on the TPU plate with 5 wt% MWCNT. A SEM image taken at very low magnification enabled us to estimate that the freeze-fractured plate has a thickness of 1.79 mm (see Figure 1a), a value that approximates the one measured by a micrometer (see Appendix A). As shown in Figure 1b,c, the carbon nanotubes are uniformly dispersed throughout the entire TPU matrix, similar to those reported in [6,38,39]. Additionally, a SEM image recorded at higher magnification (see Figure 1c,d) allowed us to detect areas where some MWCNTs are in contact with each other, creating a conductive network that could promote electronic conduction along the polymer matrix [20,40].

Figure 2 displays the XRD patterns of the studied TPU/MWCNT plates. For all samples, the diffractograms exhibit a wide main peak around 20° that is produced by the (110) planes of the TPU soft segments, in a short-ordered range, into the amorphous phase of the polymer [6,36,41]. The TPU matrices also produce a weak shoulder close to the base of the main peak, which is associated with the (002) planes of the hard segments of the polymer. In addition, the XRD patterns show peaks produced by the carbon nanotubes: a very wide peak at 43.1° (a typical graphite behavior) related to the (100) plane, and a small shoulder in the main peak at 24.4° produced by the (002) planes [42,43]. The increase in the MWCNT content helps to increase the ordered range of the TPU soft and hard segments within the amorphous matrix. They promote a well-defined peak at the top of the main peak and the formation of the small shoulder at 26.2° related to the hard segments [6,17,43].

### 2.2. Electrical Characterization

An electrical characterization of the TPU/MWCNT plates was performed through room-temperature impedance spectroscopy (IS) experiments using a two-electrode electrochemical cell configuration. Figure 3 shows the Nyquist plots (a complex-impedance plane representation of the real (Z′) and negative imaginary (-Z″) components of the impedance measured at multiple frequencies) for the plates under investigation. We see that all the IS spectra have a deformed semi-circle shape that is well defined by the MWCNT-reinforced TPU plates with 3 and 5 wt%, and partially formed for those with 1 and 7 wt%. To extract electrical information from the IS experiments, each spectrum was modeled assuming an equivalent circuit consisting in a series of a resistor (*R*_0_) and two parallel resistor-CPE (see inset of Figure 3a–c), with CPE as a constant-phase element that simulates an “imperfect” capacitor [44]. Only the IS spectrum for the TPU plate with 7 wt% MWCNT did not require the use of two parallel resistor-CPE circuit elements (see inset of Figure 3d). Although this series arrangement is not the unique equivalent circuit that could follow the same frequency-dependent impedance observed in our nanocomposites (for instance, a parallel arrangement of resistor, CPE and series resistor-CPE is also possible [45,46,47]), it provided the best fit and was consistent with what was expected according to previous works in TPU/(MW)CNT nanocomposites. The selected equivalent circuit reveals that the TPU/MWCNT plates exhibited at least three different conduction mechanisms that are represented by the resistors *R*_0_, *R*_T_ and *R*_P_, with conductivity values σ_0_, σ_T_ and σ_P_, respectively. Here, the conductivity of each resistor element was calculated through the equation σ=L/RA (*L* and *A* are the thickness and the cross-sectional area of the cylinder cell), which is plotted in Figure 4. All the electrical parameters estimated by the equivalent circuit modeling, as well as the calculated conductivities for each resistor element, are listed in Appendix A.

As shown in Figure 4, *R*_0_ exhibits the highest conductivity (σ_0_ ~ 0.01 S/cm), and it is almost constant for 1, 3 and 5 wt% MWCNT loading (for 7 wt%, the error value is greater than the adjusted *R*_0_; thus, we have omitted it), showing that its value does not depend on the MWCNT content, such that it is caused by the resistance of the Au electrodes and wire [35]. The conductivity associated with *R*_T_ increases around two orders of magnitude (from 3.2 × 10^−6^ to 2.3 × 10^−4^ S/cm) when the MWCNT content is increased from 1 to 3 wt%, maintaining constant up to 5 wt%. Finally, the conductivity of *R*_P_, which dominates the low-frequency conduction processes, follows a logarithmic increase from 1.4 × 10^−8^ to 1.1 × 10^−2^ S/cm, an improvement in the conductivity of six orders of magnitude. σ_P_ coincides with the DC conductivity (σ_DC_) of the nanocomposites, an electrical parameter that is commonly reported in graphene- and carbon-nanotubes-reinforced TPU nanocomposites [6,16,17,19,21,48]. In this work, σ_DC_ was estimated by applying the Jonscher’s power law [49] in plots of AC conductivity as a function of angular frequency, ω, for all nanocomposites under study (see Appendix A). σ_P_ (or σ_DC_) follows a similar behavior to those reported in TPU/MWCNT nanocomposites where an electronic conduction inside an insulating polymer is caused by a percolation process through the conductive network created by the carbon nanotubes. If we compare the value of σ_P_ for 1 wt% with that reported for the pure TPU, there is a significant increase in more than four orders of magnitude, so that the percolation threshold for our system should occur for a MWCNT loading lower than 1 wt%, an expected result [6,19,30]. In addition, the σ_DC_ value obtained for the maximum MWCNT content (7 wt%) is higher than those reported for TPU/CNT nanocomposites with similar weight fraction [8,16,19,20,29,31], and it is only surpassed with the use of a higher MWCNT content (25 wt%) [31], or by employing much longer CNTs (>200 μm) [16]. On the other hand, σ_T_ should be associated with a tunneling resistance where the electrons travel between non-contact carbon nanotubes separated by a small gap; its values are within the range of conductivity predicted by numerical simulation or modeling approach for CNTs (between 10^−6^ and 10^−1^ S/cm) [33,37,50,51] and follow a similar CNT volume fraction (or weight fraction) dependence. Experimentally, an electronic tunneling process is easily observed in insulate polymers with a low CNT loading (above the percolation threshold) [51,52], and it can be modeled through a parallel *R*-CPE element circuit [30,37,53].

Finally, an interesting behavior is observed in the IS spectra of the TPU plate with 5 and 7 wt% MWCNT content, where the Nyquist plots show a horizontal tail at low frequency. This behavior indicates that neither a diffusion nor blocking process is experienced by the charge carriers when they reach the metallic electrodes, so that the electrons must be the unique charge carriers present in the nanocomposites, discarding the fact that our TPU/MWCNT nanocomposites present a mixed ionic–electronic transport mechanism.

As a complementary electrical characterization, current–voltage (*I*–*V*) experiments were performed in the TPU/MWCNT nanocomposites that, due to a limitation to measure small currents, presented high electrical conductivities (TPU plates with 5 and 7 wt% MWCNT loading). As shown in Figure 5, the use of logarithm scales in both the *I*- and *V*-axes allowed us to identify at least two linear tendencies in both curves: in the low voltage region, both systems show linear behavior with a slope close to 1 (*I*~*V*), a characteristic value for an ohmic conduction mechanism; in the high voltage region, ohmic behavior is still present in the TPU plate with 7 wt% in MWCNT, while the system with 5 wt% presents a slope of 1.6, indicating the presence of a non-ohmic conduction. Comparing these results with those found in the IS experiments, the ohmic behavior of the TPU/MWCNT plate with 7 wt% is associated with the percolation conduction mechanism that seems to dominate the electrical behavior of the nanocomposite; however, for a TPU/MWCNT plate with 5 wt%, the non-ohmic behavior observed at the high-voltage region can be associated with competition of ohmic conduction by the electron percolation process and the trap-limits space charge conduction in those regions where a tunneling injection process occurs, taking into consideration that tunneling conduction can be transformed into space charge conduction when the excitation voltage is increased [54,55].

### 2.3. Shore A Hardness Test

Figure 6 shows the Shore A hardness behavior of TPU plates with different MWCNT loadings. According to the supplier company (Merquinsa-Lubrizol), the TPU Pearlthane^®^ D15N70 is manufactured with a Shore 72 A hardness. In this work, three processes occurred to produce the TPU/MWCNT plates: extrusion, pelletization and compression molding. With this in mind, we found that such a processing route provoked a reduction of the TPU hardness, especially when a uniaxial compression is used [56]. A similar hardness reduction by a low carbon-nanotube loading has also been observed on 3D-printed TPU/CNT composites [29]. However, it was found that the incorporation of nanotubes into the processed TPU plates progressively increased the Shore A hardness from 58 (1 wt%) to 68 (5 wt%) where it seemed to stabilize. This progressive hardening agrees with others works performed in similar TPU-based nanocomposites [28,36,57,58,59], and it is favored by a good dispersion of the embedded MWCNTs, demonstrating the expected strengthening of the TPU plates by the incorporation of MWCNTs. According to Shore hardness scales that we can find everywhere, our TPU/MWCNT nanocomposites can be classified as a “Medium Hard” material (similar to the tire tread) with low hardness and high elasticity.

## 3. Materials and Methods

### 3.1. Materials

The TPU/MWCNT nanocomposite plates were produced using thermoplastic polyurethane (TPU Pearlthane D15 N70) from Merquinsa-Lubrizol, Spain, and multi-walled carbon nanotubes (NC7000TM) from NANOCYL^®^, Belgium. MWCNTs have a 90% carbon purity, an average length of 1.5 μm with an average diameter of 9.5 nm.

### 3.2. Fabrication of TPU/MWCNT Composite Plates

The TPU/MWCNT nanocomposites containing 1, 3, 5 and 7 wt% MWCNT were produced using a co-rotating twin screw extruder (COPERION W&P ZSK25), varying some operating parameters (screw speed, throughput, and screw profile), as described by Benedito et. al. [15] in order to obtain the mechanical energy needed to ensure the appropriate MWCNT dispersion. The extruded filaments were pelletized using a strand pelletizer that produces pellets of about 3 mm length. To obtain the plates, pellets with different weight fraction of MWCNTs were processed using compression molding in a hot platen LabPro400 Fontijne Press (Holland) using stainless steel frames. The compression molding was carried out at 200 °C, applying 10 kN of compression force, for 5 min, to produce plates of about 1.8 mm thickness.

### 3.3. Characterization Techniques

#### 3.3.1. Cryo-SEM Studies

A cross-sectional cryo-SEM sample of a representative TPU/MWCNT plate was prepared in a Nova-Nanolab-200 Dual Beam instrument from FEI upgraded with a cryo-setup model PPT2000, which enables substrate temperatures down to −155 °C. After cooling and transferring under vacuum conditions, the sample was placed in a preparation chamber. In the loading station, the sample was positioned, under liquid nitrogen, in the transfer shuttle that was introduced into the microscope vacuum. SEM images were recorded at different magnifications using secondary electrons accelerated at 10 kV.

#### 3.3.2. X-ray Diffraction (XRD) Studies

XRD patterns of the TPU/MWCNT plates were obtained using a MultiPurpose X-ray Diffractometer PANalytical X’Pert Pro. A platinum sample holder and Ni-filtered CuKα radiation were used. The patterns were recorded at room temperature using a 2θ range of 5° to 70°, with a step size of 0.02°.

#### 3.3.3. Impedance Spectroscopy Studies

A Wayner Kerr precision impedance analyzer in combination with an Agilent 16451b dielectric test fixture was used to perform electrical characterization of the TPU/MWCNT plates by impedance spectroscopy (IS) experiments. The plates were cut into a circular shape of about 8.7 mm diameter and placed between to Au electrodes, forming a two-electrode electrochemical cell. The impedance data were collected in the frequency range of 20 Hz to 5 MHz, using an excitation signal of 100 mV. All measurements were performed at room temperature.

#### 3.3.4. Current–Voltage Experiments

The current–voltage (*I*–*V*) experiments were measured in air, and at room temperature, by a four-probe method using a Sensor-CASSY 524-013 coupled energy source meter. For this test, we used the same dielectric test fixture employed in the IS experiments.

#### 3.3.5. Shore A Hardness Studies

The hardness test of the plates was performed using a Shore A durometer (Albuquerque industrial, New York, NY, USA), according to ASTM D 2240. The plates were cut into square shapes of 10 cm size. Ten points were tested for each sample to obtain an average, and the distance between two points was 10 mm.

## 4. Conclusions

In this study, the incorporation of MWCNTs progressively improved the degree of crystallinity, electronic conductivity and hardness properties of TPU plates fabricated by combining extrusion, pelletization, and compression molding processes. All these improvements were favored by the good dispersion of the MWCNTs embedded into the TPU polymer matrix, even using a high weight fraction (7 wt%). The XRD analysis showed that the increase in the MWCNT loadings helped to increase the ordered range of both soft and hard segments. According to [22,60], the increase in the degree of crystallinity of the polymer matrix favors the rigidity of the plates and prevents any ionic conduction, something that was confirmed by the electrical and hardness characterizations. The potential of IS analysis, combined with equivalent circuit modeling, helped to determine that the TPU/MWCNT nanocomposites only exhibited electron conductivity through two conduction mechanisms: (i) percolation conduction via the conductive network created by carbon nanotubes that present contact resistance between them and (ii) tunneling conduction between nearby nanotubes that have a small gap. The percolation conduction, which was observed at a low-frequency regime, exhibited a logarithmic tendency with respect to the MWCNT loading, reaching a maximum conductivity value of ~1 × 10^−2^ S/cm for the maximum weight fraction studied here; the tunneling conduction, which always exhibited an electron conductivity higher than the percolation one, was clearly observed in the plates with low MWCNT weight fraction, and it seemed to collapse at 5 wt%, where σ_T_ and σ_P_ are very close. This result seems to indicate that, for our studied system, the percolation conduction starts to dominate the electrical behavior when the MWCNT content is higher than 5 wt%, an interesting behavior corroborated by the current–voltage experiment and that gives us the possibility to tune the electrical behavior of the conducting polymer by changing the content of the MWCNTs. Finally, the hardness experiments showed that MWCNT addition also benefited the Shore A hardness of the TPU plate, reaching a maximum value at 5 wt% loading.

## Figures and Tables

**Figure 1 molecules-28-03598-f001:**
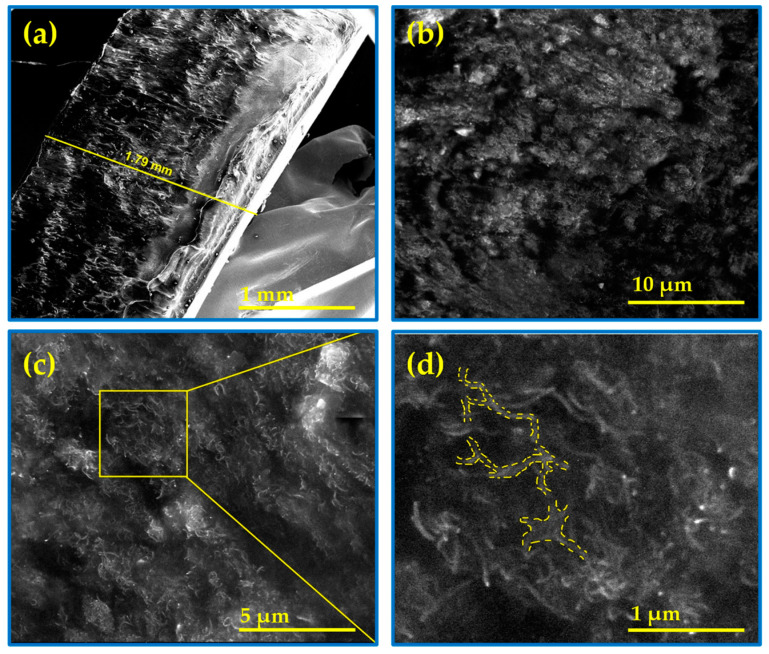
Cryo-SEM images taken on the TPU/MWCNT plate with 5 wt%. (**a**) A cross-section image recorded on the freeze-fractured plate. (**b**,**c**) are SEM images recorded at higher resolution to see the carbon nanotubes. (**d**) A zoomed image extracted from (**c**) helps to distinguish some pathways of the conductive network created by the MWCNTs along the polymer matrix.

**Figure 2 molecules-28-03598-f002:**
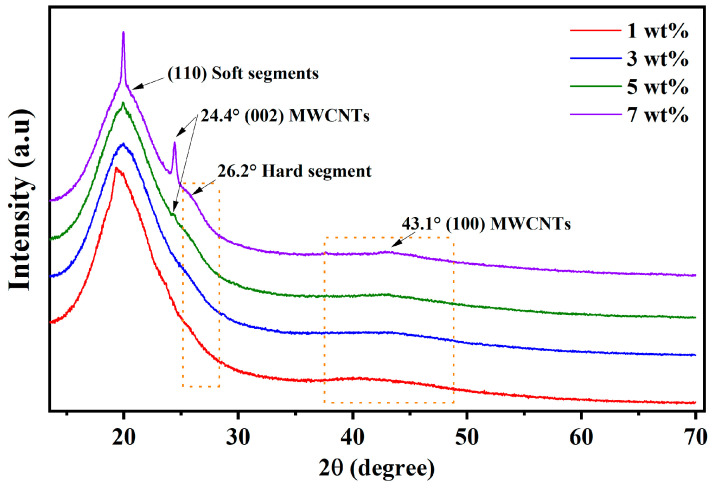
XRD patterns of the TPU/MWCNT composite plates.

**Figure 3 molecules-28-03598-f003:**
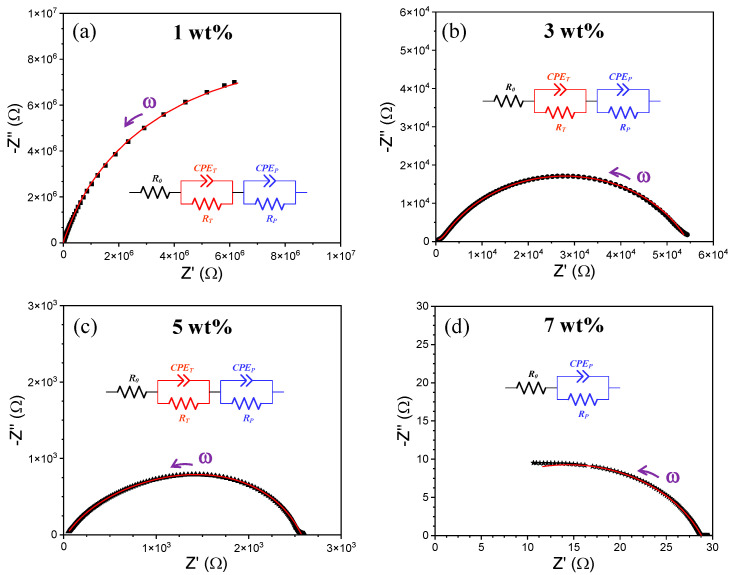
Nyquist plots of the TPU nanocomposite plates with (**a**) 1, (**b**) 3, (**c**) 5 and (**d**) 7 wt% MWCNT loading. Inside the plots are depicted the equivalent circuit that models each spectrum. Red lines represent the fit with the respective equivalent circuit. The purple arrows indicate the variation direction of the angular frequency (ω).

**Figure 4 molecules-28-03598-f004:**
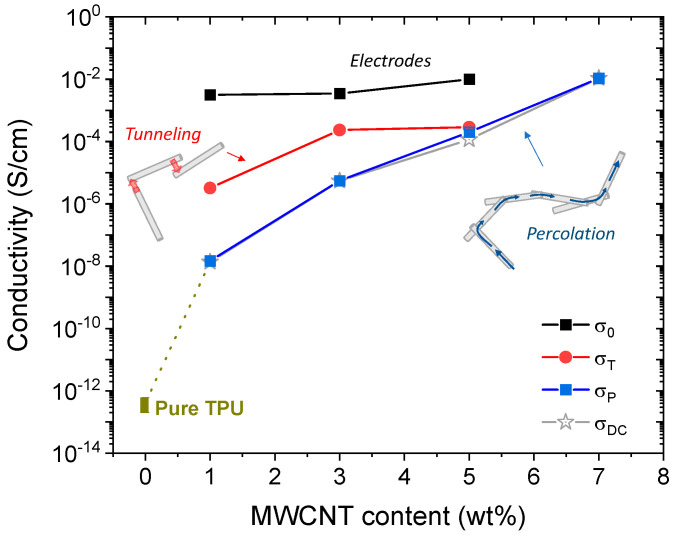
Plots of the conductivity (σ_0_, σ_T_ and σ_P_) as a function of the MWCNT content. The conductivity value for the pure TPU was extracted from [17,19] (the dot size for pure TPU represents the range of conductivity values).

**Figure 5 molecules-28-03598-f005:**
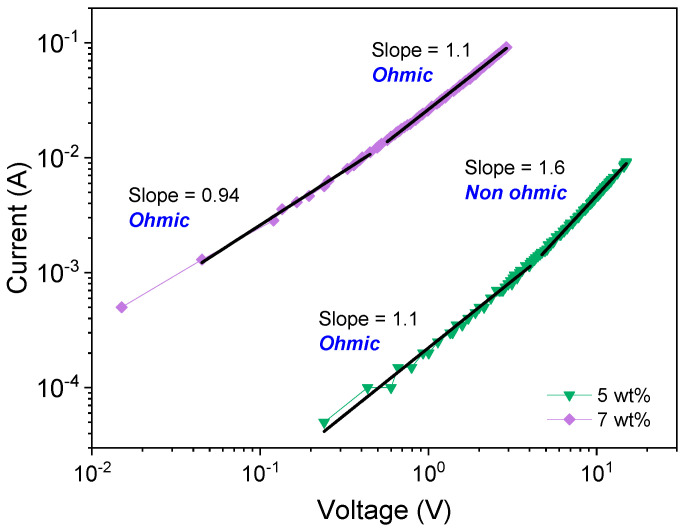
Current–voltage curves taken on the TPU/MWCNT nanocomposite with 5 and 7 wt%.

**Figure 6 molecules-28-03598-f006:**
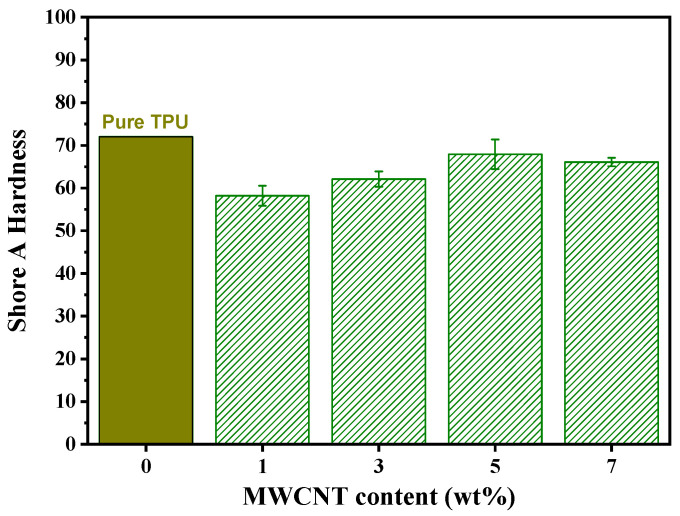
Shore A hardness behavior of TPU/MWCNT nanocomposite plates.

## Data Availability

Data is contained within the article and Appendix A.

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
