# Peer review of "Morphological Electrical and Hardness Characterization of Carbon Nanotube-Reinforced Thermoplastic Polyurethane (TPU) Nanocomposite Plates"

_molecules, 2023, doi:10.3390/molecules28083598_

Round 1

Reviewer 1 Report

The manuscript concerns with electrical characterization of carbon nanotubes nanocomposites with thermoplastic polyurethane matrix.

Electrical properties of polymeric materials containing conductive nanofillers has been intensively studied in the recent decades. Unfortunately, the manuscript reviewed does not bring a significant novel finding in this field. The authors have performed several routine electrical measurements, however, their treatment of the data obtained lacks critical and careful evaluation. Morphological analysis and hardness measurements do not contribute to the explanation or discussion of the electrical behaviour of the composites.

One of the main flaws of the manuscript is insufficiently detailed experimental part. No information is given about the polymer used as a composite matrix. Moreover, the way of preparation of the composites is unclear. It seems that the materials, i.e. TPU pellets and CNT powder, were just dry mixed and compression moulded. If this was the case, it is far away from any relevant application. Generally, the aim is to introduce as less as possible of a conductive nanofiller in order to avoid processing difficulties connected with a high viscosity increase and appearance of flow restrictions. From this point of view the motivation for the investigations high above percolation threshold is unclear.

Another crucial deficiency of the manuscript are missing experiments on a reference material, in this case on unfilled polymer matrix. This fact disables proper discussion of the results obtained from XRD, hardness and also electrical measurements. For example, the authors claim that there is ionic and/or protonic conductivity in the composites (again it is not clear from the text), but it is not discussed at all where it comes from and what kind of ions could be involved. Both mechanisms are rather rare in polymeric systems without a special modification.

The authors performed impedance spectroscopy measurements and the data were fitted by equivalent circuits. From these fits the values of ionic and electronic conductivities were obtained. However, for every particular CNT concentration another model was used without any obvious principle or rule. Thus, it is doubtful, whether these values are reasonable or not and how robust is this approach, i.e. how the conductivity values change by choosing different equivalent circuit model.

Although hardness measurements are very simple, they do not allow to draw any conclusion about general mechanical performance of the composites (strength, toughness) as the authors do. In particular in the systems under study with a high amount of the filler, I assume that the materials were rather brittle and weak.

There are also some minor flaws, which, however, show insufficient thoroughness and diligence of the authors by manuscript preparation – wrong caption of Fig. 4, missing Fig. 7 mentioned in the text.

According to the comments above I recommend the rejection of the manuscript.

Author Response

Dear reviewer 1, in the attached file you will find the answer

Reviewer 2 Report

I have read the manuscript by the authors and I believe clarifications should be made and some points must be corrected. The introduction is too short and relevant works should be added. Also, some questions must be responded to regarding this work. Finally, the supporting information the authors refer to was not uploaded, but I believe it is needed. Please see the pdf attached for all the comments.

Author Response

Dear reviewer 2, in the attached file you will find the answer

Round 2

Reviewer 1 Report

First of all, I appreciate the effort of the authors, which they put in the revision of the paper. The manuscript has been revised and rewritten significantly and, in my opinion, its scientific quality is now much higher than that of the originally submitted one. In particular, the section describing and discussing the electrical behaviour of the nanocomposites, which is the core of the paper, has been thoroughly modified and now it brings some interesting insights. Although I have still certain doubts concerning the overall novelty of the work, I think it contains some findings worth of publishing.

Reviewer 2 Report

The authors have made a great effort to improve this manuscript. I believe that this manuscript has improved a lot. The authors responded to my questions and made a lot of corrections and clarifications.